# Next-Generation Sequencing for Cystic Fibrosis: Florida Newborn Screening Experience

**DOI:** 10.3390/ijns11040094

**Published:** 2025-10-14

**Authors:** Deanna M. Green, Jean Polasky, Mark Weatherly, Heather Stalker, Colleen Blanchard, Cheryl Kushner, Marisa Couluris, Patricia Ryland, Iruvanti Sunitha, Joseph Fong, Sandra Crump, Emily Reeves, Kristin Barnette

**Affiliations:** 1Division of Pediatric Pulmonology and Cystic Fibrosis, Johns Hopkins All Children’s Hospital, St. Petersburg, FL 33701, USA; jpolask2@jhmi.edu; 2Division of Pulmonology and Sleep Medicine, Orlando Health Arnold Palmer Hospital for Children, Orlando, FL 32806, USA; mark.weatherly@orlandohealth.com; 3Division of Pediatric Genetics and Metabolism, University of Florida, Gainesville, FL 32608, USA; stalkhj@peds.ufl.edu; 4Coastal Genetic Services, St. Augustine, FL 32080, USA; colleen@coastalgeneticservices.com; 5Pulmonary and Sleep Center, Joe DiMaggio Children’s Hospital, Hollywood, FL 33021, USA; ckushner@mhs.net; 6Division of Pediatric Pulmonology, University of South Florida, Tampa, FL 33612, USA; mcouluri@usf.edu; 7Newborn Screening Laboratory, Florida Department of Health, Bureau of Public Health Laboratory, Jacksonville, FL 32202, USA; patricia.ryland@flhealth.gov (P.R.); iruvanti.sunitha@flhealth.gov (I.S.); joseph.fong@flhealth.gov (J.F.); 8Newborn Screening Follow-Up Program, Florida Department of Health, Tallahassee, FL 32399, USA; sandra.crump@flhealth.gov (S.C.); emily.reeves@flhealth.gov (E.R.); kristin.barnette@flhealth.gov (K.B.)

**Keywords:** cystic fibrosis, quality improvement, newborn screening, sweat test, immunoreactive trypsinogen, next generation sequence

## Abstract

Cystic fibrosis (CF) is an autosomal recessive genetic condition affecting nearly 1 in 4000 newborns. Early diagnosis and treatment have been shown to improve the care of individuals with CF, which is enhanced through newborn screening (NBS). The state of Florida has been performing CF NBS since 2007, and in 2022, Florida implemented enhanced next generation sequencing (NGS). The goal of this change was to identify individuals from under-represented racial and ethnic groups, who may have rare or de novo variants. NBS screening for CF involved a first tier with immunoreactive trypsinogen (IRT) ≥ 50 or the top 4% of daily specimens, whichever is lower, reflexing to a second tier. As of 2022, the second tier has evolved to an expanded sequence with an Agena 74-variant panel. Single variants would then reflex to the third tier utilizing NGS. NGS is able to confirm what is detected in second-tier testing, adding variants not included in the Agena panel, and refining the TG replications for Poly-T variants to determine pathogenicity of 5T results. When there is a variant of varying clinical consequence between the two databases, the most conservative classification is selected. Individuals with variants would then be referred to one of the contracted CF NBS referral centers for confirmatory sweat chloride testing (sweat). With implementation of NGS, referrals nearly tripled in 2022–2024, with 538 referrals in 2019; 485 in 2020; and 805 in 2021; followed by 1223 referrals made in 2022; 1146 in 2023; and 1294 in 2024. In 2022–2024, 71% of referrals to the contracted NBS CF referral centers were for single variant results, and no cases of CF were identified from these referrals. The number of CF cases remained about the same, ranging from 23 to 40 through the years 2019–2024. The number of CRMS/CFSPID cases, however, tripled going from 10 to 12 in 2019–2022 to over 100 in 2024. The reason for this change seems to be related to complex heterozygous genetic variants as opposed to abnormal sweat. Implementation of NGS for CF in Florida led to a significant increase in the identification of *CFTR* variants which affected all aspects of the NBS CF process, from an increased workload on the NBS laboratory and follow-up staff, to an increase in referrals to the NBS CF referral centers. The majority of referrals were for single-variant results, which meant the infants had a very low likelihood of having CF. It is recommended that when an algorithm involving NGS is utilized, one should verify that there are appropriate processes for sweat, including the manner in which single-variant CF results are handled, avoiding unnecessary healthcare utilization.

## 1. Background and Significance

Newborn screening (NBS) has been conducted across the United States (US) for more than 60 years, with screening for cystic fibrosis (CF) available since 1979 [1]. CF is an autosomal recessive genetic condition affecting nearly 1 in 4000 newborns in the US but varies by ethnicity and region [2]. Early diagnosis and treatment have been shown to improve the care of individuals with CF [3,4], which can be further enhanced through screening at birth.

Every state in the US screens for CF; however, each state performs this screening differently [5]. All state screening algorithms include immunoreactive trypsinogen (IRT) as the first-tier test. Second- and third-tier tests employ genetic-based techniques for the detection of CF-causing variants, with different numbers of detectable variants screened in each state. Targeted *CFTR* deoxyribonucleic acid (DNA) panels are used in multiple states, and, in a few states, next generation sequencing (NGS) has been implemented. CF can occur in people of all ethnicities and races. Limited DNA panels are less likely to identify individuals from varied racial and ethnic groups, who may have rare variants [6,7].

Florida began screening for CF in 2007 using a two-tiered methodology; IRT for the first tier, and a selective DNA panel for the second tier. Every infant with a CF-causing variant detected was referred to one of the 11 CF NBS referral centers contracted by the Florida Department of Health (FDOH). These centers perform diagnostic evaluation, including sweat chloride (sweat) testing, obtaining a family history, and providing genetic counseling to families. The final diagnostic information is reported back to the NBS Follow-up Program.

Florida has a multi-ethnic population with significant genetic admixture, which made it important for FDOH to provide a more robust evaluation of the state’s newborns. As such, in November 2021, the second-tier selective DNA panel was expanded from 60 to 74 variants. Then, in January 2022, Florida implemented enhanced next generation sequencing (NGS), with the variant library fully open to allow for detection of all CF-causing variants as a third-tier test. All exonic and some intronic CF-causing variants in CFTR2 (https://cftr2.org) (accessed on 23 September 2025) [8] and ClinVar (https://www.ncbi.nlm.nih.gov/clinvar/?term=%22cftr%22%5BGENE%5D&redir=gene) (accessed on 23 September 2025) [9], listed as pathogenic, likely pathogenic, a variant of varying clinical consequence (VVCC), and/or a variant of uncertain significance (VUS), are reported. Historically, any infant identified through NBS with at least one variant would be referred to one of the NBS CF referral centers within the state for diagnostic evaluation.

As has been documented by other states utilizing more extensive genetics, an unintended consequence of enhanced NBS, especially with NGS, is the identification of individuals with an inconclusive diagnosis [10,11]. This uncertain designation may be due to variants which are not adequately classified as pathogenic, or due to sweat resulting in an indeterminant range. In the US, this designation is termed cystic fibrosis transmembrane regulator (CFTR)–related metabolic syndrome (CRMS) or, in other parts of the world, CF screen positive, inconclusive diagnosis (CFSPID) [12]. Further complicating NBS is that individuals with a single variant are classified as a carrier. In populations screened for CF with NGS, it is possible to have more individuals classified as CRMS/CFSPID and carriers than CF. However, the intention of NBS is to identify people at risk for a particular condition to provide life-altering interventions, not to identify people who are carrying traits for a condition. The purpose of this report is to describe Florida’s lessons learned and quality improvement efforts after the implementation of NGS for CF.

## 2. Methods

### 2.1. NBS CF Screening Algorithm

CF screening in Florida has gone through multiple iterations specifically in the second tier. The first tier has remained unchanged with IRT ≥ 50 or the top 4% of daily specimens, whichever is lower, reflexing to the second tier. From 2007 to 2016, the Hologic DNA panel was used for second-tier testing. From 2016 to 2021, it was the Luminex 60 variant panel. As mentioned, in November 2021, an expanded DNA panel from Agena with 74 variants was adopted for Tier 2. Included in this expanded panel were poly-thymidine (Poly-T) variants; however, the panel did not provide thymidine-guanine (TG) evaluation, which is important for pathogenicity. It was known prior to implementation that a large percentage of the population had a Poly T variant; however, the exact incidence rate of this in Florida was unknown.

In January 2022, NGS was added as the third tier in Florida’s CF screening process, which provides the full gene sequence of the exonic regions and some intronic regions. CFTR dele 2,3 was also included in the panel as large deletions are common in Florida’s population. NGS is able to confirm what is detected in second-tier testing, adding variants not included in the Agena panel, and refining the TG replications for Poly-T variants to determine the pathogenicity of 5T results. Since the NBS Follow-up algorithm required referral for any detected variant, many infants were referred that would not have been had the TG replication been known. The current FL algorithm is provided in Figure 1. Of note, those with CF borderline in Tier 2 were sent for Tier 3 testing. Those with CF borderline in Tier 3 were referred to the CF referral centers for further diagnostic testing.

All 1085 variants annotated in CFTR2.org [8] in 2024, those in ClinVar [9] and those in gnomAD (CFTR|gnomAD v4.1.0|gnomAD) [13], were reportable. When there is a variant of varying clinical consequence between the two databases, the most conservative classification is selected. For example, a variant identified as benign in CFTR2.org but VUS in ClinVar would be reported as a VUS. Individuals with any of these combinations would then be referred to one of the contracted CF NBS referral centers for confirmatory sweat.

### 2.2. Overall Design

This is a cohort study of infants born in Florida from 2019 to 2024 who underwent newborn screening blood spot testing and were referred to a contracted NBS CF referral center for an out-of-range screening result during these time periods. Detection rates of at least one CFTR variant in the CF NBS population pre-NGS (2019–2021) and post-NGS (2022–2024) implementation were compared. Variant reporting was provided and annotated by the state NBS laboratory. All data are presented as descriptive and categorical.

## 3. Results

### 3.1. CF NBS Referrals

In the six years evaluated, there were 1,309,130 infants screened by the Florida NBS Program. In the three years preceding the expanded DNA panel and NGS, the number of referrals to the 11 contracted NBS CF centers was 538 in 2019, 485 in 2020, and 805 in 2021. The referral number nearly doubled from 2020 to 2021 due to algorithm changes with the expanded number of detectable variants and inclusion of Poly-T variants in the DNA panel. In the 10.5 months from January through mid-November 2021, 507 infants were referred for CF diagnostic evaluation. From mid-November through December 2021, 305 infants were referred, largely due to detection of Poly-T variants with unknown TG replication. With the implementation of NGS in 2022, referrals for Poly-T variants were limited to those with a TG replication of ≥12; however, referrals were still drastically increased due to the number of additional variants being identified on the NGS panel (Figure 2). Despite the increased number of referrals, the number of confirmed CF diagnoses remained stable or decreased, while designations of carriers and CRMS/CFSPID rose substantially (Table 1).

Additionally, NGS found multiple compound heterozygotes whereas years prior did not report on these. In 2022, 4 individuals were found to have six variants: 1 with five variants, 15 with four variants, and 153 with three variants. In 2023, there were 1 individual with six variants; 3 with five variants; 16 with four variants; and 175 with three variants. In 2024, there were 1 individual with five variants, 14 with four variants, and 102 with three variants (Figure 3). The most common variants with more than 30 variants reported in 2022–2024 are presented in Table 2.

### 3.2. Diagnosis of Referrals

Frequency-of-sweat Cl testing nearly tripled after implementation of NGS; however, the number of CF cases identified did not increase during these years. Of the 80 individuals with CF in 2022–2024, 1 case had four variants identified, and 2 cases had three variants. All other cases had only two pathogenic variants identified. Two individuals diagnosed with CF had normal sweat Cl, and one individual had intermediate sweat. The number of CRMS cases also increased during this time. The driving force for this change seemed to be complex heterozygous variants as opposed to abnormal sweat. Table 3 shows the differentiation of these individuals based on the number of variants detected.

## 4. Discussion

From the time that screening for CF was introduced, the goal has been timely and efficient diagnosis of individuals with CF to help alter the infant’s clinical trajectory. It is important for CF NBS protocols to cover a wide array of variants in order to fully evaluate a population. However, identifying potential cases should not be at the expense of saturating a system so that true-positive CF cases experience delays in care. Implementation of NGS will allow for increased variant detection without significantly altering true case detection. In Florida from 2022 to 2024, 71% of referrals to the contracted NBS CF referral centers were for single-variant results and no cases of CF were identified from these referrals (Figure 3).

The addition of NGS led to the identification of individuals with multiple variants and very complex compound heterozygotes. These individuals entered the medical system for further evaluation, including a sweat test. Sweat is used to confirm a diagnosis of CF; however, intermediate results lead to an inconclusive diagnosis, termed CRMS/CFSPID [12]. This scenario is exactly what Florida experienced with the addition of NGS, with triple the number of CRMS/CFSPID cases classified. This increased population is in alignment with data from Wisconsin, New York, and California, who have been using extended sequencing for identification of variants in more varied populations [10,11,14]. In Florida, intermediate sweat was only found in very few individuals (*n* = 40) following NGS implementation, and of these, 40% had only one variant. Therefore, the predominant determination for a designation of CRMS/CFSPID was genetic indecisiveness due to inclusion of VUS and/or VCCC in two or more variants. In the years prior to NGS, individuals with intermediate sweat were found to have a second variant and confirmed to have CF (five in 2019, eight in 2020, and 7 in 2021). This was not the case in the years after NGS.

Having the appropriate infrastructure in place prior to implementing NGS is paramount. Performance of sweat at high volumes can tax a medical system as this testing requires specific training and a unique, labor-intensive skill set [15]. Introducing individuals who are unlikely to have CF saturates this system and reduces availability of this highly skilled work force. When volumes rise, so can rates of Quantity Not Sufficient (QNS) results, which then requires the patient to return for a retest at a later date [16]. This stressor was noted significantly in Florida in 2022 as the accredited sweat Cl testing labs at the contracted NBS CF referral centers were not prepared for the rapid increase in referrals. Many of the centers utilized laboratory personnel which were at reduced staffing models after the COVID-19 pandemic, and many CF referral centers did not have the necessary infrastructure to provide this highly skilled testing in large volumes. The volume of referrals impacting scheduling availability, then QNS results, delayed diagnosis further, which caused significant anxiety for families as it remained unclear whether their child had CF or not [17].

Furthermore, interpretation of complex genetics will be necessary when multiple variants are found in the same individual. This requires expertise from genetic counselors (GC), geneticists, and CF clinicians working in collaboration with one another. CF clinicians’ background knowledge of when symptoms may appear in infants can assist GCs when educating families about VUS. GCs with expertise in CF genetics in turn can help ease parental anxiety [18]. In Florida, every contracted referral center is required to provide genetic counseling to families. The CF Foundation (CFF) has also recently recognized this group as necessary to CF center function [19].

Continued clinical monitoring of individuals diagnosed with CRMS/CFSPID will be necessary. Further refining of genetic variants will be necessary to determine final risk of any given variant. Inclusion of individuals diagnosed with CRMS/CFSPID in the CF patient registry will be critical [20].

Despite best efforts, CFTR sequencing protocols will inevitably miss cases of CF. For every CF NBS screening protocol, the first step is IRT elevation. There are multiple reports of individuals with CF who did not have elevated IRT at birth [21,22,23]. NGS cannot prevent this issue since it is a later step in a tiered system. Inevitably, there will be older individuals presenting with symptoms and diagnosed with CF even if they were initially screened via NBS. Since this three-tiered system has been implemented in Florida, there has not been a false negative case identified in a child under age 3.

Due to the stressors on the contracted NBS CF referral center and NBS Follow-up Program infrastructure, a work group was formed among these stakeholders in September 2023 to review data and evaluate how to address the increased volume of referrals. These stakeholders included three physicians, two genetic counselors, and two NBS coordinators from NBS CF referral centers, as well as NBS laboratory and follow-up staff. During the next year, the workgroup met to review the CF process and reach a consensus which could be shared with all 11 contracted NBS CF referral centers. This collaborative workgroup proposed only referring infants with two or more variants, and no longer referring infants with only a single variant detected. The workgroup met with colleagues from the California and New York Newborn Screening Programs to glean lessons learned from their process change. The group also met with representatives from the CFF who endorsed the proposed process changes after reviewing available data, which were further corroborated by the CFF’s NBS guideline [24]. After presenting these data as well as additional data, all of the NBS CF referral centers agreed to the process change, which was implemented in January 2025.

When only a single CF variant is identified through newborn screening, parents and the identified primary care provider (PCP) will be notified via letter of the out-of-range screening result and that the infant is most likely a carrier. These letters include a CF Single-Variant Fact Sheet which includes information on what single variant results mean, answers to frequently asked questions, and the need for genetic counseling and family planning (Appendix A). This process is similar to the one already employed in Florida for sickle cell trait (carrier) results. Through mid-April 2025, 222 single variant cases were handled using this new process. During this same time period, only 76 infants required referral for multiple-variant results. While NBS CF referral centers anticipate receiving community referrals from PCPs for single-variant results, the overall volume of infants requiring sweat testing is still likely to decrease significantly. Florida will be able to monitor for false negatives from these community referrals. Stakeholders also believe this will lead to improved parental satisfaction due to fewer families being contacted for diagnostic testing, as well as the provision of direct information to families instead of through a contracted NBS CF referral center with whom the family had no previous relationship. Another benefit of this process change is the direct education to PCPs regarding CF genetics and what the results mean for their patient, which has historically been a challenge. Genetic counseling continues to be available to families with single variants identified through referral from the PCP.

In conclusion, implementation of NGS for CF in Florida led to a significant increase in the identification of *CFTR* variants which affected all aspects of the NBS CF process, from increased workload on the NBS laboratory and follow-up staff, to an increase in referrals to the NBS CF referral centers, which therefore increased the number of patients requiring sweat. The majority of referrals were for single variant results, which meant the infants had a very low likelihood of being positive for CF. While waiting to complete confirmatory testing, many families expressed anxiety about whether or not their infant had CF. For states implementing NGS, it is recommended that an algorithm is implemented to promote appropriate referrals for sweat Cl testing, including the manner in which single-variant CF results are handled, avoiding unnecessary healthcare utilization and undue burden on families and the NBS CF system.

## Figures and Tables

**Figure 1 IJNS-11-00094-f001:**
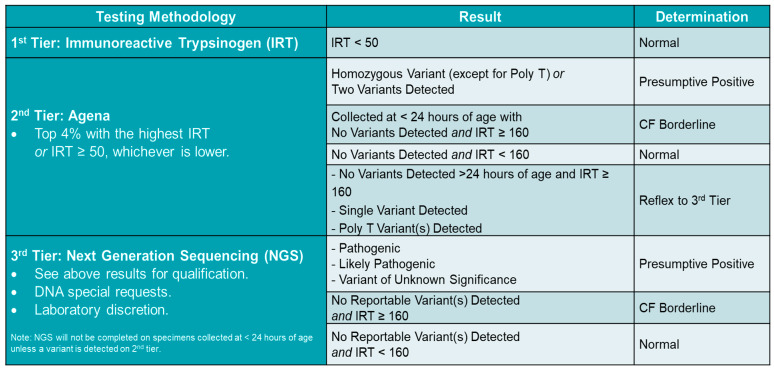
Current CF NBS algorithm.

**Figure 2 IJNS-11-00094-f002:**
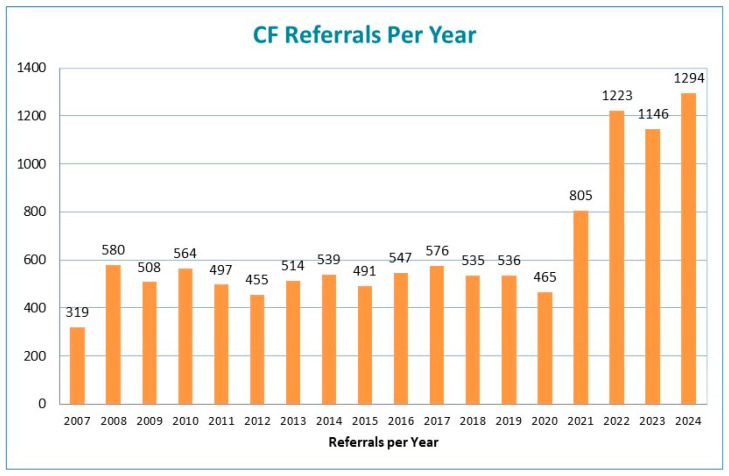
Annual referrals to contracted NBS CF centers from 2007 to 2024.

**Figure 3 IJNS-11-00094-f003:**
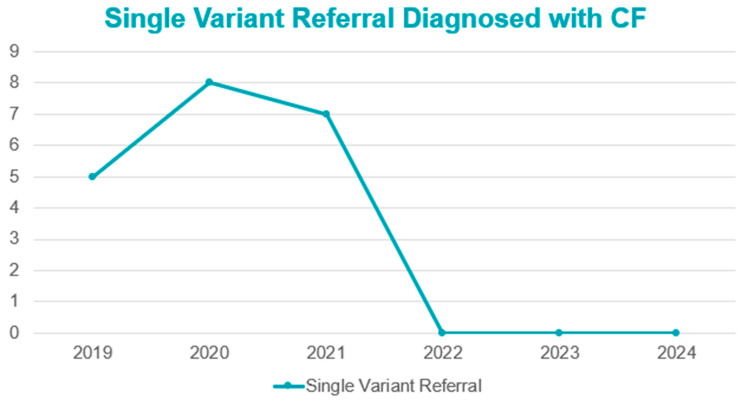
Rates of referrals for a single variant later identified as an individual with CF.

**Table 1 IJNS-11-00094-t001:** Diagnostic outcomes: CF, carriers, and CRMS/CFSPID designations.

Year	CF	CRMS/CFSPID	Carriers
2019	40	10	410
2020	38	12	335
2021	40	12	570
2022	25	54	890
2023	23	80	838
2024 *	32	106	852

* Note: 191 cases in 2024 remain pending final status.

**Table 2 IJNS-11-00094-t002:** Most frequent variants with more than 30 variants reported via NGS in 2022–2024.

	2022	2023	2024
F508del	320	235	390
PolyT-T5-TG12	298	365	367
PolyT-T5-TG13	39	21	31
R117H	65	31	51
F508C	35	32	57
3120 + 1G > A	29	22	33
R1162L	23	14	33
R75Q	16	30	64
T966T	16	25	34
I506V	18	12	33

**Table 3 IJNS-11-00094-t003:** CRMS/CFSPID cases with associated compound heterozygous state and sweat chloride classification.

Number of Variants	2022 CRMS Cases	2022 Sweat Cl Classification	2023 CRMS Cases	2023 Sweat Cl Classification	2024 CRMS Cases	2024 Sweat Cl Classification
6 variants	2	1 intermediate, 1 normal	1	1 normal	0	-
5 variants	0	-	1	1 normal	1	1 intermediate
4 variants	5	5 normal	7	2 intermediate, 5 normal	4	4 normal
3 variants	20	7 intermediate, 13 normal	28	7 intermediate, 21 normal	43	5 intermediate, 35 normal, 3 QNS
2 variants	20	2 intermediate, 18 normal	35	6 intermediate, 26 normal, 3 not performed	48	7 intermediate, 38 normal, 1 positive, 1 QNS
1 variant	7	5 intermediate, 1 normal, 1 QNS	8	2 intermediate, 3 normal, 2 not performed	10	6 intermediate, 4 normal
Total CRMS cases	54 (15 intermediate, 36 normal, 1 QNS, 2 not performed)		80 (17 intermediate, 55 normal, 1 QNS, 5 not performed)		106 (19 intermediate, 82 normal, 4 QNS, 1 positive)	

## Data Availability

Restrictions apply to the availability of these data. Data were obtained from Florida Department of Health and are available Emily Reeves and Kristin Barnette with the permission of Florida Department of Health.

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
