# Peer review of "Next-Generation Sequencing for Cystic Fibrosis: Florida Newborn Screening Experience"

_2409-515X, 2025, doi:10.3390/ijns11040094_

Round 1
Reviewer 1 Report
Comments and Suggestions for Authors
Your approach of wanting to detect and report variants of varying clinical consequences (VVCCs), and especially variants of uncertain significance (VUSs), is surprising. It results in a significant number of CRMS/CFSPID diagnoses. What is the exact ratio between these diagnoses and those of cystic fibrosis (CF)?
Furthermore, clinicians often lack the resources and time to properly interpret these variants. This work is then delegated to the screening laboratory, which represents a substantial workload.
Finally, are primary care provider sufficiently trained to communicate a heterozygosity result to the parents of a screened newborn? The information must be conveyed with great clarity so that parents understand the importance of informing other family members of this result.
Author Response
Thank you for taking the time to review our report. We agree with your comments that reporting all variants of varying clinical consequences (VVCCs) and variants of uncertain significance (VUS) would lead to a significant number of more CRMS/CFSPID diagnoses (comment 1). Helping other groups understand that making a change to Next Generation sequencing could lead to many more diagnoses of CRMS/CFSPID should you open the library to all variants is exact reasoning behind wanting to submit this report. We had attempted to address this concern in Table 1, Page 6. As mentioned there the ratio of cases of CRMS/CFSPID greatly increased in 2023 and 2024. The ratio would be 2023 in 3.47:1 and in 2024 it would be 3.31. We have not provided this in Table 1 but could add a column addressing this if you feel it would make it more easily interpreted.
Thank you for additionally pointing out that normally the work of determining variates will be delegated to the screening laboratory instead of the clinicians (comment 2). In our state the screening laboratory has been provided the list of variants to be reported out as pathogenic, likely pathogenic, variant of uncertain significance or benign. The laboratory staff does not make any judgement calls. These reports are provided to the Cystic Fibrosis referral centers and the clinicians there make the final determination and are the ones to finally interpret these variants and provide appropriate counseling to families.
We do agree with your comment 3 that primary care provider (PCP) need to be sufficiently trained to communicate a heterozygosity result to the parents of a screened newborn. As discussed in the report, our state provides letters to the PCP and facts sheets. We are continually working on further educational resources to aid with training of providers. For better clarity of the information provided, we can include this information in supplementary material. See reference to this material on Page 8 line 259. We do agree this will be of great interest to other groups changing their screening algorithms.
Reviewer 2 Report
Comments and Suggestions for Authors
This article provides new data on the real-world impact of implementing next generation sequencing (NGS) in Florida’s cystic fibrosis newborn screening program. Its novelty lies in showing how NGS increased detection of complex genotypes and CRMS/CFSPID cases, while not raising the number of confirmed CF diagnoses. The study also documents the strain on laboratories and referral centers caused by higher referral volumes. However, the workflow and screening algorithm are not clearly explained. In addition, the management pathway for CF borderline determinations should be described in more detail.
This manuscript reports Florida’s experience implementing next generation sequencing (NGS) in cystic fibrosis (CF) newborn screening between 2019 and 2024. The study compares pre- and post-NGS referral and diagnostic outcomes, highlighting a marked increase in referrals, stable CF diagnoses, and a tripling of CRMS/CFSPID designations. The authors discuss the impact on laboratory workload, clinical capacity for sweat chloride testing, and parental anxiety. Policy changes in 2025 to reduce unnecessary referrals are also described.
Comments
- Line 23–24: The prevalence estimate of “one in 4,000 newborns” is somewhat outdated and varies across regions and ethnic groups. A recent reference with specific data should be provided.
- Line 27–28: The phrase “de novo variants” is misleading in the CF context, since CFTR mutations are almost always inherited. The term should be removed or clarified with citations.
- Line 196–198: Carriers with intermediate sweat chloride may reflect technical artifacts, modifier gene effects, or undetected second variants. The authors should avoid over-calling these simply as carriers.
- Line 230–236: “IRT will identify true cases about 95–99% of the time” is not real because sensitivity depends on race/ethnicity, the popultaion genotypes and on IRT cutoff values. False negatives are well documented, particularly in non-ΔF508 genotypes and in certain ethnic groups.
Methods: Explain more clearly the workflow or screening algorithm and provide a detailed description of how borderline CF determinations (figure 1) are managed.
Author Response
- Thank you for taking the time to review our manuscript. We agree with your comment 1 that prevalence of CF does vary across regions and ethnic groups. Scotet and colleagues as recent as 2020 reassessed this prevalence and continues to state this is 1 in 4,000 newborns in the US. We have clarified our paper to note that we are referring specifically to prevalence in the US but that this varies by region and ethnicity as noted and includes this reference. See page 1 line 59-60 which now reads “CF is an autosomal recessive genetic condition affecting nearly 1 in 4,000 newborns in the US but varies by ethnicity and region (2).”
- Thank you for pointing out “de novo variants” is misleading. We agree with your comment and have removed this language. Line 71 just reads who may have rare variants.
- Thank you for addressing the reasons for false positive tests. We agree that we may have been over simplifying to refer to these individuals as carriers. We have edited this section lines 197-199 to state only our data with no inferences. “In Florida, intermediate sweat was only found in very few individuals (n=40) following NGS implementation and of these 40% had only one variant.”
- While we agree that true sensitivity will depend on race/ethnicity and population studied, we chose to include a reference for this statement. We will remove this statement as the rest of the paragraph is our point, that false negative will still exist if they were never identified as an abnormal IRT.
- Thank you for this insightful question regarding our methods. In Figure 1 CF borderline which are those collected at <24 hours of age with no variants detected and IRT>160 are sent on to 3rd Tier testing. If there are still no reportable variants detected and IRT >160 after 3rd Tier the current algorithm involves sending these reports to the CF referral center for sweat testing and genetic counseling. It is then at the discretion of the CF referral center to make a final diagnosis. We have clarified this adding line 123-125 which reads “Of note, those with CF borderline in Tier 2 are sent for Tier 3 testing. Those with CF borderline in Tier 3 are referred to the CF referral centers for further diagnostic testing.”
- We have also revised Table 3 for slightly better readability. We believe this is the table or figure that could be clearer. We will request of the editor to make sure it is all on one page for better understanding